# Not too little, not too much:
# a theoretical analysis of graph (over)smoothing

**Nicolas Keriven**
CNRS, GIPSA-lab
`nicolas.keriven@cnrs.fr`

## Abstract

We analyze graph smoothing with *mean aggregation*, where each node successively receives the average of the features of its neighbors. Indeed, it has been observed that Graph Neural Networks (GNNs), which generally follow some variant of Message-Passing (MP) with repeated aggregation, may be subject to the *oversmoothing* phenomenon: by performing too many rounds of MP, the node features tend to converge to a non-informative limit. At the other end of the spectrum, it is intuitively obvious that *some* MP rounds are necessary, but existing analyses do not exhibit both phenomena at once. In this paper, we consider simplified linear GNNs, and rigorously analyze two examples of random graphs for which a finite number of mean aggregation steps provably improves the learning performance, before oversmoothing kicks in. We identify two key phenomena: graph smoothing shrinks non-principal directions in the data faster than principal ones, which is useful for regression, and shrinks nodes within communities faster than they collapse together, which improves classification.

## 1 Introduction

In recent years, deep architectures such as Graph Neural Networks (GNNs), along with the availability of large sets of graph data, have significantly broadened the field of machine learning on graphs and structured data, see [2, 3, 8, 19] for reviews. Most GNNs rely on the **Message-Passing** (MP) framework [7, 11]. At each layer $k$, for each node $i$, a representation $z_i^{(k)}$ is computed using the representations of the *neighbors* $\mathcal{N}_i$ of $i$ in the graph at the previous layer: $z_i^{(k)} = \text{AGG}\left(\{z_j^{(k-1)}\}_{j \in \mathcal{N}_i}\right)$, where AGG is an **aggregation function** that treats $\{z_j^{(k-1)}\}_{j \in \mathcal{N}_i}$ as an *unordered set*, to respect the absence of node ordering in the graph. Here we consider one of the most classical, *mean aggregation*:

$$z_i^{(k)} = \frac{1}{\sum_j a_{ij}} \sum_j a_{ij} \Psi\left(z_j^{(k-1)}\right) \tag{1}$$

where the $a_{ij} \in \mathbb{R}_+$ are the entries of the adjacency matrix of the graph, and $\Psi$ is some function (usually a Multi-Layer Perceptron).While MP is a natural and rather general framework, its limitations were quickly observed by researchers and practitioners. Foremost among them is the so-called *oversmoothing* phenomenon [14]: as the GNN gets deeper and many rounds of MP are performed, the node features $z_i^{(k)}$ tend to become too similar across the graph. To relieve it, researchers have explored residual mechanisms [6, 13], dropping connections [9], clever normalizations [21] or regularizations [5], among others. Some works have acknowledged the important role of the aggregation function, and proposed new exotic diffusion strategies [1] or to optimize it [12].

On the theoretical side, oversmoothing has mostly been analyzed in the infinite-layer limit $k \to \infty$. In this case, classical spectral analysis of graph operators such as the Laplacian can be leveraged to indeed show that node features will always converge to some limit that carries a limited amount of information [17]. This is particularly true for mean aggregation (1). However, there has been little research at the other end of the spectrum. Generally, researchers show the power of GNNs

N. Keriven, Not too little, not too much: a theoretical analysis of graph (over)smoothing (Extended Abstract). Presented at the First Learning on Graphs Conference (LoG 2022), Virtual Event, December 9–12, 2022.

for a *sufficient* (unbounded) number of layers, such as the now-famous ability to distinguish graph isomorphism as well as the Weisfeiler-Lehman test and all its variants [16, 20]. Since these results are valid for an unbounded number of layers, the settings adopted in these works are, by definition, incompatible with non-informative oversmoothing.

In a recent preprint [10][1], we showcase two representative exemples, of regression and classification, on which *linear* GNNs (sometimes called SGC [18]) are provably subject to this double phenomenon: *some* smoothing is useful for learning, while *too much* smoothing inevitably leads to oversmoothing.

We adopt on a model of latent space random graphs, with node features. We identify two key phenomena for this: smoothing shrinks non-principal directions in the data faster than principal ones (Sec. 4), and shrinks communities faster than they collapse together (Sec. 5). Although our theoretical settings are obviously simplified, we believe it is a step towards a better comprehension of graph aggregation, of the relationship between node features and graph structure. All proofs are given in the full paper [10], of which the present document is an extended abstract.

## 2 Preliminaries

**Notations.** The norm $\|\cdot\|$ is the Euclidean norm for vectors and spectral norm for (rectangular) matrices. For a psd matrix $\Sigma$, the Mahalanobis norm is $\|x\|_{\Sigma}^2 \stackrel{\text{def.}}{=} x^\top \Sigma x$. The determinant of a matrix is $|S|$, and its smallest eigenvalue is $\lambda_{\min}(S)$. The multivariate Gaussian distribution with mean $\mu$ and covariance $\Sigma$ is denoted by $\mathcal{N}_{\mu,\Sigma}(x) = \det(2\pi\Sigma)^{-\frac{1}{2}} e^{-\frac{1}{2}\|x-\mu\|_{\Sigma^{-1}}^2}$.

**SSL..** In this paper, we consider Semi-Supervised Learning (SSL) [4, 11] on an undirected graph of size $n$. We observe a *weighted* adjacency matrix $A = [a_{ij}]_{i,j=1}^n \in \mathbb{R}_+^{n \times n}$ as well as *node features* $z_1, \ldots z_n \in \mathbb{R}^p$ at each node of the graph. We also observe *some* labels $y_1, \ldots, y_{n_{\text{tr}}} \in \mathbb{R}$ at training time and aim to predict the remaining labels $y_{n_{\text{tr}}+1}, \ldots, y_n$. For simplicity, we assume that $n_{\text{tr}}$ and $n_{\text{te}}$ are both in $\mathcal{O}(n)$. We denote by $Z \in \mathbb{R}^{n \times p}$ the matrix whose rows contain the node features, $Z_{\text{tr}}, Z_{\text{te}}$ respectively its first $n_{\text{tr}}$ and last $n_{\text{te}}$ rows, and similarly $Y_{\text{tr}}, Y_{\text{te}}$ the vectors containing the observed and non-observed labels.

**Graph smoothing with mean aggregation.** Here we consider a simplified situation of *linear* GNN with mean aggregation, often used as a theoretical baseline [18]. A linear GNN with $k$ layers just corresponds to performing $k$ rounds of mean aggregation on the node features, then learning on the smoothed features. We denote by $d_A = [\sum_i a_{ij}]_j \in \mathbb{R}_+^n$ the vector containing the degrees of the graph and $D = \text{diag}(d_A)$. Assuming that all degrees are non-zero, performing one round of mean aggregation corresponds to multiplying $Z$ by $L = D^{-1}A$. The smoothed node features after $k$ rounds of mean aggregation are: $Z^{(k)} = L^k Z$. Each row, denoted by $z_i^{(k)} \in \mathbb{R}^p$, contains the smoothed features of an individual node. Its first $n_{\text{tr}}$ and last $n_{\text{te}}$ rows are denoted $Z_{\text{tr}}^{(k)}, Z_{\text{te}}^{(k)}$.

**Learning.** In this paper, we consider learning with a Mean Square Error (MSE) loss and Ridge regularization. For $\lambda > 0$, the regression coefficients vector on the smoothed features is

$$\hat{\beta}^{(k)} \stackrel{\text{def.}}{=} \text{argmin}_\beta \frac{1}{n_{\text{tr}}} \left\| Y_{\text{tr}} - Z_{\text{tr}}^{(k)} \beta \right\|^2 + \lambda \|\beta\|^2 = \left( \frac{(Z_{\text{tr}}^{(k)})^\top Z_{\text{tr}}^{(k)}}{n_{\text{tr}}} + \lambda \text{Id} \right)^{-1} \frac{(Z_{\text{tr}}^{(k)})^\top Y_{\text{tr}}}{n_{\text{tr}}} \quad (2)$$

Then, the test risk is defined as

$$\mathcal{R}^{(k)} \stackrel{\text{def.}}{=} n_{\text{te}}^{-1} \left\| Y_{\text{te}} - \hat{Y_{\text{te}}}^{(k)} \right\|^2 \quad \text{where } \hat{Y_{\text{te}}}^{(k)} = Z_{\text{te}}^{(k)} \hat{\beta}^{(k)} \quad (3)$$

Our goal is to illustrate some situations where a finite amount of smoothing provably improves the test risk, that is, there is an optimal $k^\star > 0$ such that $\mathcal{R}^{(k^\star)} < \min(\mathcal{R}^{(0)}, \mathcal{R}^{(\infty)})$.

**Random graph model.** We adopt popular *latent space random graph models* akin to graphons [15]. In these models, to each node $i$ is associated an *unobserved latent variable* $x_i \in \mathbb{R}^d$ with $d \geqslant p$, and edge weights are assumed to be equal to $a_{ij} = W(x_i, x_j)$ where $W : \mathbb{R}^d \times \mathbb{R}^d \to \mathbb{R}_+$ is a *connectivity kernel*. Note that edges may also be taken as *random Bernoulli variables*, but we do not consider this here for simplicity. Moreover, we consider that the $(x_i, y_i)$ are drawn *iid* from

---

[1]the reference has been anonymized for review purpose.

some joint distribution, and the node features are a linear projection of the latent variables to a lower dimension: $z_i = M^\top x_i$ for some unknown $M \in \mathbb{R}^{d \times p}$ that satisfies $M^\top M = \mathrm{Id}_p$. To summarize:

$$\forall i, j, \quad (x_i, y_i) \overset{iid}{\sim} P, \quad z_i = M^\top x_i, \quad a_{ij} = W(x_i, x_j) \tag{4}$$

For this model, note that $Z^{(k)} = X^{(k)} M$ where $X^{(k)} = L^k X$. In the rest of the paper, we use the Gaussian kernel with a small additive term $\varepsilon > 0$:

$$W(x, y) = \varepsilon + W_g(x, y) \quad \text{where } W_g(x, y) \overset{\text{def.}}{=} e^{-\frac{1}{2}\|x-y\|^2} \tag{5}$$

The coefficient $\varepsilon$ is added to lower-bound the degrees of the graph and avoid degenerate situations.

# 3 Oversmoothing

In this section, we briefly examine the oversmoothing case, when $k \to \infty$ while all other parameters are fixed. In this case, it is well-known that all node features converge even for general GNNs [17]. For completeness, we state below this result in our settings. We denote by $\bar{d} = d_A / d_A^\top 1_n$ the vector of normalized degrees, which is also the limit distribution of a random walk on the graph.

**Theorem 1.** *Define $v = Z^\top \bar{d}$ and $\bar{y}_{\mathrm{tr}} = n_{\mathrm{tr}}^{-1} \sum_{i=1}^{n_{\mathrm{tr}}} y_i$. We have $\hat{Y}_{\mathrm{te}}^{(k)} \xrightarrow[k \to \infty]{} \left( \frac{\|v\|^2}{\lambda + \|v\|^2} \bar{y}_{\mathrm{tr}} \right) 1_{n_{\mathrm{te}}}$.*

Hence, in the limit $k \to \infty$, the predicted labels become all equal. Using simple concentration inequalities, it is generally easy to show that $\mathcal{R}^{(\infty)} \approx \mathrm{Var}(y) + \mathcal{O}(1/\sqrt{n})$ when $\lambda \to 0$. In most cases, this leads to situations where $\mathcal{R}^{(0)} < \mathcal{R}^{(\infty)}$, and oversmoothing occurs.

# 4 Finite smoothing: Linear Regression

In this section, we consider a problem of linear regression on Gaussian data. We consider $x \sim \mathcal{N}_{0,\Sigma}$ for some positive definite covariance matrix $\Sigma$, and $y = x^\top \beta^\star$, without noise for simplicity. For a symmetric positive semi-definite matrix $S \in \mathbb{R}^{d \times d}$, we define the following function

$$R_{\mathrm{reg.}}(S) \overset{\text{def.}}{=} (\Sigma^{\frac{1}{2}} \beta^\star)^\top \left( \mathrm{Id} - S^{\frac{1}{2}} M (\lambda \mathrm{Id} + M^\top S M)^{-1} M^\top S^{\frac{1}{2}} \right)^2 (\Sigma^{\frac{1}{2}} \beta^\star) \in \mathbb{R}_+ \tag{6}$$

where we recall that $M$ is the projection matrix to obtain the node features $z = M^\top x$. Note that it satisfies $0 \leqslant R(S) \leqslant \|\beta^\star\|_\Sigma^2$. We additionally define $\Sigma^{(k)} = (\mathrm{Id} + \Sigma^{-1})^{-2k} \Sigma$.

**Theorem 2** (Regression risk without smoothing.)**.** *With probability at least $1 - \rho$,*

$$\mathcal{R}^{(0)} = R_{\mathrm{reg.}}(\Sigma) + \mathcal{O}\left( \frac{\|\Sigma\| \|\beta^\star\|^2 d \sqrt{\log(1/\rho)}}{(\lambda + \lambda_{\min}^{(0)}) \sqrt{n}} \right) \tag{7}$$

$$\mathcal{R}^{(1)} = R_{\mathrm{reg.}}(\Sigma^{(1)}) + \mathcal{O}\left( C \varepsilon^{1/5} \right) + \mathcal{O}\left( \frac{C' \log n \sqrt{d + \log(1/\rho)}}{(\lambda + \lambda_{\min}^{(1)}) \sqrt{n}} \right) \tag{8}$$

*where $C = \mathrm{poly}(\|\Sigma\|, e^d, |\mathrm{Id} + \Sigma|)$, $C' = \mathrm{poly}(\varepsilon^{-1}, \|\Sigma\|, \|\beta^\star\|)$ and $\lambda_{\min}^{(k)} = \lambda_{\min}(M^\top \Sigma^{(k)} M)$.*

This theorem gives a limiting expression of $\mathcal{R}^{(0)}$ and $\mathcal{R}^{(1)}$ with additional error terms. Since it is easy to show that for $n$ large enough $\mathcal{R}^{(0)} \leqslant \mathcal{R}^{(\infty)} \approx \mathrm{Var}(y)$, we obtain the following result.

**Corollary 1.** *Take any $\rho > 0$, and suppose $R_{\mathrm{reg.}}(\Sigma^{(1)}) < R_{\mathrm{reg.}}(\Sigma)$. If $\varepsilon$ is sufficiently small and $n$ is sufficiently large, then with probability $1 - \rho$, there is $k^\star > 0$ such that $\mathcal{R}^{(k^\star)} < \min(\mathcal{R}^{(0)}, \mathcal{R}^{(\infty)})$.*

In other words, under some hypothesis on $R_{\mathrm{reg.}}$, there is indeed coexistence of beneficial finite smoothing and oversmoothing. Below we exhibit a simple example where this hypothesis is satisfied.

As expected in linear regression, the *covariance* of the $x_k^{(k)}$ is key in the expression of the risk. It can be seen in the proof of Theorem 2 (available at [10]) that $x^{(1)}$ behaves like $(\mathrm{Id} + \Sigma^{-1})^{-1} x$, whose covariance is $\Sigma^{(1)}$, hence the consequence that $\mathcal{R}^{(1)} \approx R_{\mathrm{reg.}}(\Sigma^{(1)})$. Similarly, by applying repeated smoothing we can extrapolate that $x^{(k)}$ behaves like $(\mathrm{Id} + \Sigma^{-1})^{-k} x$, such that $\mathcal{R}^{(k)} \approx R_{\mathrm{reg.}}(\Sigma^{(k)})$. The matrix $\Sigma^{(k)}$ has the same eigendecomposition as $\Sigma$, but where every $\lambda_i$ is replaced by $\lambda_i^{(k)} =$

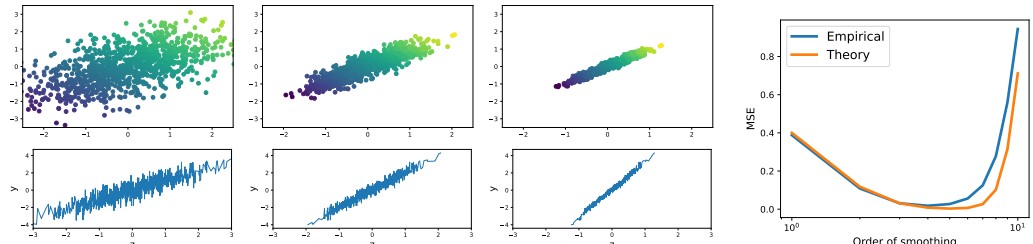

**Figure 1:** Illustration of mean aggregation smoothing on the 2D example described in the text. **First three figures on the left, top:** *unobserved* latent variables $X^{(k)}$ in dimension $d = 2$ where the colors are the $Y$; **bottom:** observed node features $Z^{(k)} = X^{(k)}M$ in dimension $p = 1$ on the x-axis, labels $Y$ on the y-axis. **From left to right**, three order of smoothing $k = 0, 1$ and $2$ are represented. **Figure on the right:** comparison of empirical and theoretical MSE (details in [10]) with respect to order of smoothing $k$.

$(1 + 1/\lambda_i)^{-2k}\lambda_i$. This can be interpreted as follows: when $\lambda_i \gg 1$ is large, $\lambda_i^{(k)} \sim \lambda_i$, while if $\lambda_i \ll 1$ is small, $\lambda_i^{(k)} \sim \lambda_i^{2k+1}$. Hence smoothing **shrinks the directions of the small eigenvalues faster than that of the large ones**. Thus, if $\beta^\star$ is mostly aligned with the eigenvectors of large eigenvalues, smoothing may *reduce unwanted noise* in the node features $z = M^\top x$.

We illustrate this on a toy situation (Fig. 1). Consider the following settings: $d = 2, p = 1$, $\Sigma$ has two eigenvalues $\lambda_1 = 2$ and $\lambda_2 = 1/2$, with respective eigenvectors $u_1 = [1, 1]/\sqrt{2}$ and $u_2 = [-1, 1]/\sqrt{2}$, and $\beta^\star = bu_1$. Finally, $M^\top = [1, 0]$ is the projection on the first coordinate. In this case, we can compute explicitly: $\mathcal{R}^{(k)} \approx R_{\text{reg.}}(\Sigma^{(k)}) = \lambda_1 b^2 \frac{(2\lambda + \lambda_2^{(k)})^2 + \lambda_2^{(k)}\lambda_1^{(k)}}{(2\lambda + \lambda_1^{(k)} + \lambda_2^{(k)})^2}$. So, if $\lambda_2^{(k)}$ decreases faster than $\lambda_1^{(k)}$, this function will first decrease to a minimum of approximately $\lambda_1 b^2 (\frac{2\lambda}{2\lambda + \lambda_1^{(k^\star)}})^2$ (when $\lambda_2^{(k)} \approx 0$), before increasing again to $\lambda_1 b^2 = \|\beta^\star\|_\Sigma^2 = \lim_{n \to \infty} \mathcal{R}^{(\infty)}$.

## 5 Finite smoothing: classification

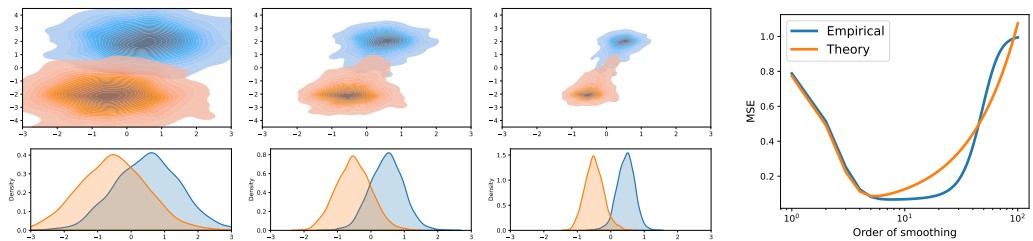

**Figure 2:** Illustration of mean aggregation smoothing on a classification task with two Gaussians with dimensions $d = 2$, $p = 1$, where $M$ projects on the first coordinate. **First three figures on the left, top:** density of *unobserved* latent variables $X^{(k)}$ in dimension $d = 2$; **bottom:** density of observed node features $Z^{(k)} = X^{(k)}M$ in dimension $p = 1$. **From left to right**, three order of smoothing $k = 0, 1$ and $2$ are represented. **Figure on the right:** comparison of empirical and theoretical MSE with respect to order of smoothing $k$.

In this last section, we examine a simple classification problem for two balanced classes with Gaussian distribution: $(x, y) \sim (1/2)(\mathcal{N}_{\mu,\text{Id}} \otimes \{1\} + \mathcal{N}_{-\mu,\text{Id}} \otimes \{-1\})$. We note that this is not a *difficult* problem *per se*, and that linear regression is certainly not the method of choice to solve it. Our main goal is to illustrate the smoothing phenomenon. Our main result is the following.

**Theorem 3.** *Take any $\rho > 0$. If $\varepsilon$ is sufficiently small, and $\|\mu\|, n$ are sufficiently large, and $\|M^\top \mu\| > 0$, then with probability $1 - \rho$, there is $k^\star > 0$ such that $\mathcal{R}^{(k^\star)} < \min(\mathcal{R}^{(0)}, \mathcal{R}^{(\infty)})$.*

Note that we have assumed $\|\mu\|$ to be sufficiently large here. However, we do *not* assume that $\|M^\top \mu\|$ is large (just non-zero), and the classification problem on the $z_i$ alone may be very difficult. As seen in the proof [10] and Fig. 2 on a $d = 2$ example, the interpretation here is the following: in the proper regime, **communities will initially concentrate in the latent space faster than they get closer from each other**, which helps learning on the $z^{(k)}$. Then, they eventually collapse together.

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
