# OpenReview forum: "Not too little, not too much: a theoretical analysis of graph (over)smoothing"
_logconference.io/LOG/2022/Conference — LoG 2022 Oral_

### Official Review · Reviewer_C5HT · 2022-09-24

**Overall Score:** 8
**Confidence:** 4

**Review:**


This paper provides a novel analysis of the over smoothing of linear GNN with mean aggregation in the nonasymptotic regime. It shows that there exists a sweet spot for depth k where the risk $R^k$ will be smaller than $min(R^0, R^{\infty})$, both in regression and classification cases.

The paper is well written with strong theoretical contributions. It is acknowledged that the setting is simplified, but given that most analysis of over smoothing is asymptotic, I think this is a nice contribution and therefore recommend acception.

In corollary 1, it is assumed that $R_{reg}(\Sigma^{(1)}) < R_{reg}(\Sigma)$ . How realistic is this assumption? Also, my understanding is that the current analysis doesn’t touch on the optimization aspect of GNN. Can you comment on the challenges of extending your analysis to nonlinear GNN?

---

### Official Review · Reviewer_6HUT · 2022-10-17

**Overall Score:** 6
**Confidence:** 4

**Review:**

**Summary**
---

This paper looks at the trade off between over smoothing and generalization error and show that under certain circumstances there is an optimal level of over smoothing. This analysis is done is a simple setting. First the GNN learns features using random walk types smoothening proof. Once we have these features a linear classifier is trained on the features. The paper looks at two situations. The first is in the completely over smoothened regime and in the limited smoothening regime. It also has some experiments to validate the theoretical claims.

**Contributions**
---

I think theorem 2 is the main contribution in which they provide the asymptotic risk rate that holds with high probability. They do this in the no smoothening and one iteration of the random walk smoothening case. Theorem 3 is just an example situation in which we have the conclusion of corollary 1 of theorem 2.

**Strengths**
---

The paper looks at a very clean and simple set up and shows an interesting result, which is that some smoothening is required.

This is interesting because the targets are generated using a linear function for each input $x$. That is, there is no graph structure here it is just a linear function without any noise. Hence one might expect that linear regression on our features (which are random projections of the true features) should be the best we can do.

However, as the paper shows, if we smoothened the features by doing a random walk on a random graph generated using a Gaussian Kernel, then we learn a linear map that has better generalization properties.

Hence I think the result is interesting. In such a situation I think the simplicity of the model is a strength and not a weakness as it shows interesting phenomena even in easy cases.

**Weaknesses**
---

1) I think the writing of the paper is the main weakness. In particular, I found the discussion of Theorem 2 (linear 108 to 124) difficult to follow.

Other issues, I do not follow the assumptions/results for Theorem 1. Specifically Theorem 1 says let $v = Z^d \bar{d}.$ However, I do not think $Z$ or $\bar{d}$ is defined anywhere in the paper. I think we can infer that $Z$ is the feature matrix in the limit with infinite smoothening. However, I do not know what $\bar{d}$ is. Other $d$s that exists in the paper are $d$ for the embedding dimension and $d_A$ for the degree vector of the graph $A$ however, I don't think these are related to $\bar{d}$.

2) Another weakness for me is the missing proofs. Maybe this is to be expected for an extended abstract. The authors do cite an anonymous version of the paper that has the proofs and is supposed to be arXiv.

3) The missing proofs + the writing is what makes it difficult to understand linear (108 to 124). For example, line 110 references a $x^{(k)}_k$ However, should this be a $z^{(k)}_k$ (based on line 69)?

Following this lines 112 and 113 claim that the risks are approximately the $R_{reg}$ terms. However, it is very unclear to me why we can ignore the other terms. Is this because in the large $n$ limit and in the small $\epsilon$ case, these are small?

**Improvement Suggestions**
---

1) I think improving the writing will help.

2) I think some intuition behind equation 6 would be helpful.

3) I understand the example at the end is meant to help and there is limited space, but some discussion on the assumption $R_{reg}(\Sigma^{(1)}) < R_{reg}(\Sigma^{(0)})$ would also help.

---

### Official Review · Reviewer_oRhh · 2022-10-20

**Overall Score:** 8
**Confidence:** 5

**Review:**

**Summary**

This work works on the theoretical analysis of the over-smoothing problem of Graph Neural Networks (GNNs). Under the linear assumption and the Mean Square Error (MSE) loss and Ridge regularization setting, the authors provide two theoretical conclusions:
1. Smoothing shrinks the directions of the small eigenvalues faster than that of the large ones.
2. Communities will initially concentrate in the latent space faster than they get closer to each other.
Although the analysis is in a simplified setting, the conclusions are aligned with the recent empirical study and heroical analysis[1,2].

The 4-page requirement definitely limits this work. I think more details should be given to make reviewers justify the correctness of those theorems. I hope the author can extend this submission to a long paper in the future.

In general, I like this work. It directly points out the focus of this analysis at the beginning which makes the work easy to follow. Besides, I think the conclusions of this submission are important and interesting. I think the eigenvalue shrinkage view is able to inspire more practical work in the future.

**Typo**

Line 56: change SSL.. —> Semi-Supervised Learning (SSL).


**Reference**

[1] Wang, Haonan, Jieyu Zhang, Qi Zhu, and Wei Huang. "Augmentation-Free Graph Contrastive Learning." arXiv preprint arXiv:2204.04874 (2022).

[2] Yuansheng Wang, Wangbin Sun, Kun Xu, Zulun Zhu, Liang Chen, and Zibin Zheng. Fastgcl: Fast self-supervised learning on graphs via contrastive neighborhood aggregation. arXiv preprint arXiv:2205.00905, 2022.

---

### Official Review · Reviewer_ggA7 · 2022-10-21

**Overall Score:** 8
**Confidence:** 4

**Review:**

***Paper Summary***

The paper presents a theoretical study of how the smoothing phenomenon affects the learning performance of linear GNNs that utilize a mean aggregation. Specifically, based on a risk minimization setup, the authors show that a value of $0<k<\infty$ layers/rounds exists that yields a risk less than the initial $R^{(0)}$ and the infinite (after a 'large' number of layers) $R^{(\infty)}$ one. Although the result is not novel (from an empirical aspect), the whole risk-based approach and the presentation of the two random graph examples is very interesting and possibly helpful for the GNN community. What I, also, can recognize as novelty is the formulation of the key phenomena about the collapse of the non-principal directions and the collapse of the nodes within communities.

***Questions***
1. The authors base their theoretical study on the case of linear GNNs (i.e. non-linear activation functions do not exist, and, thus, the sequential applications of linear GNN layers can collapse to a single product one). It would be very helpful, also, having an insight on how non-linear GNNs (which is mostly often the case in applications) are affected from over-smoothing under this framework. Could the authors provide some intuition on whether an how their theory could be extended to non-linear activation functions?
2. Except for the linearity of the GNNs, the authors, also, make the assumption of a mean aggregation operator. Can the authors discuss on possible extension of the risk-based theoretical results for general choices of aggregators? Such a discussion could reveal some hindsights on desired properties of aggregation operators for the design of deeper GNNs.

***Evaluation***

Overall, I think that the present paper reveals interesting information on how the over-smoothing impacts the behavior of linear GNNs and I recommend its acceptance.

---

### Meta-Review · Area_Chair_iSjn · 2022-11-16

**Confidence:** 5
**Recommendation:** Accept for spotlight

**Meta Review:**

This is an extended abstract that investigates theoretically the issue of over-smoothing in the context of graph neural networks. All reviewers felt that this constitutes a novel and potentially important theoretical contribution to the graph learning community, which I agree based on my own reading of the paper. The results on the existence of a finite number of layers provably leading to improvement of learning performance is novel, and the interpretation of what graph smoothing does in regression and classification settings is interesting. The results of the paper may shed light on further research in the area. Therefore I recommend accept for spotlight.

---

### Decision · Program_Chairs · 2022-11-22

Accept (Oral)